# HiDF: A Human-Indistinguishable Deepfake Dataset

## ABSTRACT

The rapid development and prevalence of generative AI has made it easy for people to create high-quality deepfake images and videos, but their abuses also have been exponentially increased. To mitigate potential social disruption, it is crucial to quickly detect authenticity of each deepfake content hidden in a sea of information. While researchers have worked on developing deep learning-based methods, the deepfake datasets utilized in these studies are far from the real world in terms of their qualities; most of the popular deepfake datasets are human distinguishable. To address this problem, we present a novel deepfake dataset, HiDF, a high-quality and human-indistinguishable deepfake dataset consisting of 30 K images and 4 K videos. HiDF is a meticulously curated dataset that includes diverse subjects, which has been undergone rigorous quality checks. Comparison on the quality between HiDF and existing deepfake datasets demonstrates that HiDF is human-indistinguishable, hence it can be used as a valuable benchmark dataset for deepfake detection tasks. Data and code (`https://github.will.be.provided`) are publicly available for future deepfake detection research.

## 1 INTRODUCTION

*DeepFake*, a compound term originated from **Deep** learning and **Fake**, refers to audio and visual content that has been manipulated or generated by artificial intelligence (AI) techniques, which can hardly distinguishable by human eyes (Masood et al., 2023; Heidari et al., 2024). With the advancements of generative AI techniques such as Generative Adversarial Networks (GANs) (Karras et al., 2019; 2020) and diffusion structures (Blattmann et al., 2023; Guo et al., 2023; Liu et al., 2024), people can easily generate high-quality DeepFake content that can make others unable to verify its authenticity. Such a capability, on one hand, has populated the generation and use of DeepFake content for diverse purposes such as in film, gaming, advertising, and entertainment, providing time and cost efficiency (Campbell et al., 2022; Usukhbayar & Homer, 2020; Murphy et al., 2023). On the other hand, DeepFake content has been increasingly used for disinformation, political propaganda, and nonconsensual sexual deepfakes (MacKenzie & Bhatt, 2020; Gosse & Burkell, 2020; Maddocks, 2020), which has caused social disruption and jeopardize ethical foundations.

The increasing importance of preventing abuse of DeepFake content has spurred research communities to develop the detection methods for DeepFake images or videos. The methods for DeepFake image detection have mostly focused on spatial differences caused by manipulation, such as local noise (Wang & Chow, 2023), artifacts (Shiohara & Yamasaki, 2022; Cao et al., 2022; Zhao et al., 2021), and fine-grained texture details (Liu et al., 2020b; Chai et al., 2020). The prior studies to detect DeepFake videos have mainly focused on discovering discrepancies between adjacent frames that occur over times (Choi et al., 2024; Gu et al., 2022; Zheng et al., 2021; Xu et al., 2024; Bonettini et al., 2021). This is followed by research on multimodal deepfake detection, which utilizes multiple modalities such as audio and visual information in detecting deepfake content. Extensive research has been conducted on the inconsistencies between the audio-visual modalities in deepfake videos, which demonstrates the effectiveness of utilizing multiple modalities rather than relying only on a single audio or visual feature in deepfake video detection (Zheng et al., 2021; Cozzolino et al., 2023; Feng et al., 2023; Ha et al., 2020; Yang et al., 2023).

So far, the deepfake image and video detection work has relied on the publicly available Deep-Fake datasets such as DFDC (Dolhansky et al., 2020), FakeAVCeleb (Khalid et al., 2021), and

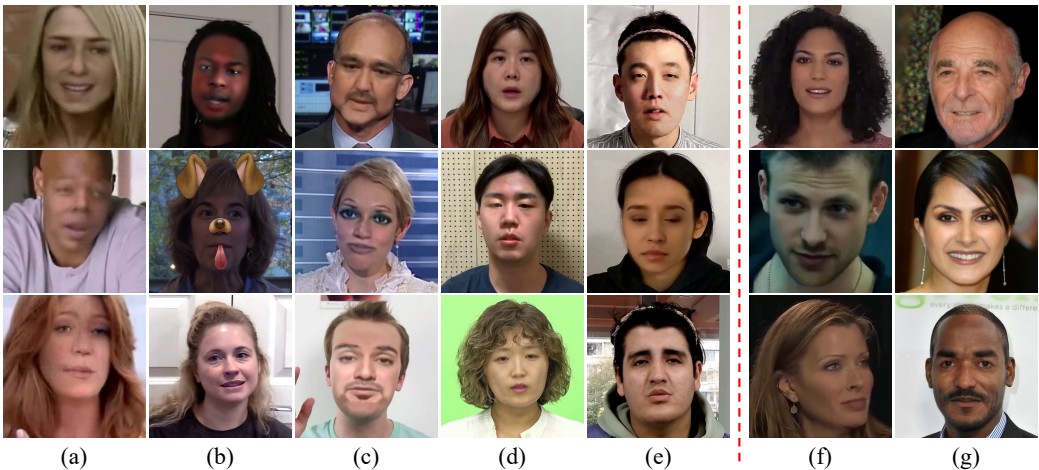

(a)      (b)      (c)      (d)      (e)      (f)      (g)

Figure 1: Samples of existing deepfake datasets and HiDF. (a) FakeAVCeleb, (b) DFDC, (c) FF++, (d) KoDF, (e) DFGC, (f) videos in HiDF, (g) images in HiDF. Figures 1a to 1f show frames extracted from videos.

FF++ (Rossler et al., 2019) – the detection models proposed in this work were built and evaluated by the public datasets. Unfortunately, these deepfake datasets contain a significant amount of visually unnatural data, with blurred edges in the synthesized parts or improperly aligned faces (Figure 1a to 1e), which can easily be recognized by human so that it is far from the current practice of DeepFake content generated by commercial deepfake applications. Since most deepfake misuse cases stem from people's inability in detecting DeepFake content, human-indistinguishable DeepFake data that can be easily created by recent commercial deepfake tools is essential to evaluate the capabilities of deep fake detection methods that can be applicable in practice.

Therefore, we propose a novel high-quality deepfake dataset called Human-indistinguishable Deep-Fake (HiDF), which includes high-quality 30 K images and 4 K videos, each of which is rigorously reviewed. Our qualitative evaluations by humans confirm that HiDF is perceived as 'more authentic' than real data, demonstrating consistent high quality. In generating deepfake data in HiDF, instead of applying well-known simple methods like FSGAN (Nirkin et al., 2019), we use the commercial tools that are widely accessible and used by the public. In this way, HiDF includes natural synthesis outcomes that are not easily distinguishable by humans. Notably, HiDF can play a role as a useful benchmark for future research that fights against realistic deepfake content generated by commercial tools.

We construct HiDF as a multimodal deepfake dataset that includes images (i.e., visual) and videos (i.e., visual and audio). Using both audio and visual modalities together is known to be effective in deepfake detection because they can capture subtle mismatches between the actual speech and the manipulated face. While most of existing deepfake datasets include unrelated audio to visible video content, e.g., just containing a voice recording of a person rather than a person who speaks in a video, we only include data where its visual and audio information is exactly matched. For ensuring generalizability, HiDF incorporates a large number of subjects.

The key contribution of this paper is summarized as follows.

- We propose HiDF, a novel high-quality multimodal deepfake dataset with 30 K images and 4 K videos that are rigorously reviewed. Our comprehensive experiments demonstrate that HiDF is human indistinguishable and comprehensive, which can be used as a valuable benchmark for future deepfake detection research. We open HiDF publicly available at `https://github.will.be.provided`.

## 2   BACKGROUND AND MOTIVATION

There have been publicly available deepfake datasets that are popularly used for deepfake detection. FaceForensics++ (FF++) (Rossler et al., 2019) comprises 1 K real videos collected from YouTube

Table 1: Quantitative comparison of HiDF and existing deepfake datasets. Real, Fake, and Total for HiDF represent the combined count of images and videos. Tool indicates whether commercial tools were used for generating the deepfake data, and Quality denotes whether a quality assessment of the dataset was performed. Q: Quantitative (using evaluation metrics such as FID, PSNR, SSIM) only, QQ: Both Quantitative and Qualitative (including pilot studies such as human surveys), N/A: Not applicable.

| Dataset | # Real | # Fake | # Total | # Subject | Data Type | Tool | Quality |
|---|---|---|---|---|---|---|---|
| FF++ | 1,000 | 4,000 | 5,000 | N/A | Image, Video (w/o Audio) | ✗ | N/A |
| ForgeryNet | 1,537,831 | 1,579,478 | 3,117,309 | $5400 + \alpha$ | Video (w/o Audio) | ✗ | N/A |
| DFDC | 23,654 | 104,500 | 128,154 | 960 | Video (w/ Audio) | ✗ | N/A |
| KoDF | 62,166 | 175,776 | 237,942 | 403 | Video (w/ Audio) | ✗ | Q |
| FakeAVCeleb | 500 | 19,500 | 20,000 | 500 | Video (w/ Audio) | ✗ | N/A |
| DFGC | 2,019 | 3,270 | 5,289 | 40 | Video (w/ Audio) | ✓ | N/A |
| UADFV | 290 | 301 | 591 | 49 | Video (w/o Audio) | ✓ | N/A |
| HiDF | 34,491 | 34,491 | 68,982 | $6,217 + \alpha$ | Image, Video (w/ Audio) | ✓ | QQ |

and 4 K fake videos manipulated from the real ones by four different synthesis methods (Thies et al., 2016; Faceswap, 2018; Deepfakes, 2018; Thies et al., 2019). The synthesized method of each fake video is also provided, which helps to develop detection models that do not depend on a single synthesized method. Note that although a large amount of deepfake images (around 1.8 M) can be obtained by the provided script that extracts individual frames from individual videos, which was used by a few studies for detection of deepfake images, the resolution of each image is too low to be used for misuse in practice. ForgeryNet (He et al., 2021) is another popular deepfake dataset, comprising 2.9 million images and 221,247 videos. It spans diverse variations, including 7 image manipulation techniques, 8 video manipulation techniques, and 36 perturbation attacks. However, similar to FF++, the manipulated content in ForgeryNet is easily distinguishable by humans, making it far less representative of real-world deepfake data. Additionally, both FF++ and ForgeryNet do not include the audio information of the videos, which disables multimodal-based approaches for deepfake detection.

More recently, deepfake datasets consisting of videos with audio have been emerged to support multimodal-based approach for deepfake detection. DFDC (Dolhansky et al., 2020) is a popular dataset that includes a number of real and fake videos. The real videos are taken with 3,426 paid actors recorded videos in natural environment without professional lighting or makeup. From the real videos, eight different synthesizing methods to swap faces of a pair of the actors were used, which resulted in the generation of more than 100 K fake videos. Despite a huge amount, a high portion of the dataset was recorded in dim lighting or extremely dark conditions where the faces in these videos are less recognizable, which is hardly feasible in practice. Furthermore, the detail information of manipulating process, such as types of manipulated data (audio or video) or synthesized methods is not provided, which can lead to model bias or overfitting.

Another high-resolution deepfake video dataset featuring Korean subjects is KoDF (Kwon et al., 2021), which includes more than 62 K real and 175 K fake videos generated by synthesizing video frames with 6 different methods (Faceswap, 2018; Perov et al., 2020; Nirkin et al., 2019; Siarohin et al., 2019; Yi et al., 2020; Prajwal et al., 2020). Although the subjects in the videos are well-balanced in terms of gender and ages, the single-race composition (i.e., Korean) limits diversity, which can also restrict the generalizability of the deepfake detection model to other races. In addition, all the real and fake videos are based on recordings of participants reading a script, so the dataset suffers from the lack of representation for moving subject.

In contrast, FakeAVCeleb seeks to support general deepfake detection in terms of race and gender. In particular, 500 celebrities across different ethnicities and genders were chosen by VoxCeleb2 (Chung et al., 2018) and used them as the subjects of real and fake videos. Three different types of manipulation (i.e., fake audio only, fake video only, and both fake audio and video) for fake videos are provided together. Despite comprehensive consideration on the dataset construction, the detailed information of (i) preprocessing such as criteria for filtering corrupted videos and (ii) qualitative/quantitative evaluations is insufficient, so the quality of the dataset can not be assured. Note that we have found a number of unnatural fake videos that can easily caught by human eyes, as shows in Figure 1a.

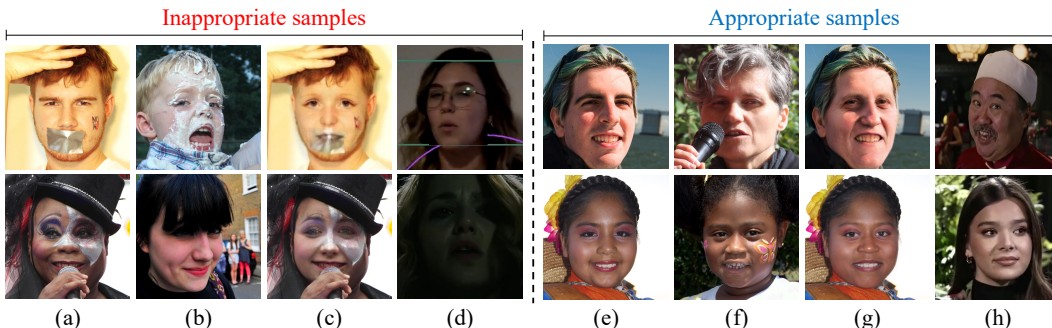

Figure 2: Examples of appropriate and inappropriate base and target images and videos. (a): Base image, (b): Target image, (c): Result of face swap with (a) and (b), (d): Base video. (a) to (d) depict examples that are unsuitable as base and target, while (e) to (h) illustrate appropriate examples that correspond to (a) to (d).

It is worth to noting that the synthesizing methods used in the existing datasets to produce fake videos is limited since the quality of the outputs by the methods is low to have a negative impact to human society. In addition, we have found that there is a non-negligible gap between the outputs by the synthesizing method and advanced commercial tools; the generated deepfake content by commercial tools is human-indistinguishable while the ones by the synthesizing methods is not. In line of this, DFGC, originating from The Second DeepFake Game Competition, has recently been proposed. The dataset contains 3,270 fake videos, of which 471 were generated by 4 commercial tools (ZAO, FaceMagic (FaceMagic), ReFace (Reface), Jiggy (Jiggy)) and a synthesizing method (YouTube-DF (Kukanov et al., 2020)). Although the quality of fake videos is similar to real-world deepfake content, the videos are still human-distinguishable since the co-provided audio is not related to the subjects in the video. Furthermore, the number of the subject across all the videos is only 40, which is insufficient to support a development of unbiased deepfake detection models. Similarly, UADFV (Yang et al., 2019) is a deepfake dataset generated by a commercial tool (i.e., FakeApp[1]), containing 49 deepfake videos and 252 images. However, like DFGC, it has a limited amount of deepfake data and subjects, restricting the depth of information.

Table 1 summarizes the quantitative comparison between HiDF and the other deepfake datasets. All the fake videos in HiDF were created by commercial tools and a significantly larger number of subjects (around 6 K) are in the videos, compared to other datasets. This ensures the provision of high-quality data with guaranteed diversity. We also conducted the comprehensive quality assessment for HiDF, which can leverage HiDF's feasibility in future deepfake detection research.

## 3    HiDF: A Human-Indistinguishable DeepFake Dataset

In this section, we introduce the construction process for HiDF. In particular, we first describe the process of selection of base images/videos, annotation, generation of deepfake content, and quality inspection. After showing the results of basic analysis, we report the ethical considerations of our data collection process, followed by the declaration of license of HiDF.

### 3.1    Methodology

To construct HiDF, we first rigorously choose (i) two types of images (base and target) to be swapped into and (ii) base videos. After annotating the information of the subject in individual images/videos, we recruited the paid applicants to manually generate fake videos and images. We finally conduct a manual inspection process to ensure the quality of the generated videos. Here, we describe each process in detail.

---

[1]FakeApp 2.2.0. https://www.malavida.com/en/soft/fakeapp/

### 3.1.1 INITIAL DATA PREPARATION

**Base and target images:** To choose high-quality base and target images, we first consider all the images in two public datasets, CelebA-HQ (Karras et al., 2017) and Flickr-Faces-HQ (FFHQ) (Karras et al., 2019). CelebA-HQ is a high-quality version of CelebA (Liu et al., 2015), providing 6,217 celebrity face images. FFHQ is an image dataset collected from the online website, comprising 70,000 images of public faces.

Based on a number of candidate images, we focus on only the images mainly featuring a face of a single person and choose base images by filtering out the images with (i) any obstructions (e.g., hair, hats) covering the eyes, nose, or mouth, and (ii) foreign substances (e.g., protruding decorations, mud) or excessive makeup (e.g., cosmetics, face painting), as illustrated in Figure 2a and 2b. For the target images to be swapped into base images, we slightly ease the conditions. In particular, we permit (i) partial obstructions of the face (ii) makeup is allowed (See Figure 2e (upper and lower, respectively)). Note that the images with foreign substances on the face are still ignored.

**Base videos:** For the selection of the base videos, we first collected the videos of celebrities from YouTube. Among the collected videos, we exclude the videos with (i) added filters (Figure 2d (upper)), (ii) excessively dark lighting (Figure 2d (lower)), (iii) rapid or excessive head movements, and (iv) no speaking subjects. The reason for the last criterion is to provide both audio and video information for multimodal deepfake detection. From the selected videos, we extract video clips to feature only one person based on motion (Bewley et al., 2016) and identity (Deng et al., 2019) to enhance the quality of the generated deepfake videos (i.e., performance of synthesis). Note that we set the length of the video clips to 3 seconds. In addition to the processed videos, we also use the real videos in FakeAVCeleb (Khalid et al., 2021) as base videos.

Throughout the process, we finally obtain 34,933 base images, 41,777 target images, and 5,707 base videos. A detailed explanation of the dataset is described in Appendix A.

### 3.1.2 ANNOTATIONS FOR RACE, GENDER, AND AGE

**Annotation criteria** Not only the images and videos, but also we provide additional information of race, gender, and age of the subject in each image/video. For race, we adopt the racial classification commonly used by the U.S. Census Bureau (White, Black, Asian, Hawaiian, and Pacific Islanders, native Americans, and Latino) as outlined by Karkkainen & Joo (2021). Considering the distinct outward differences, we detach Indians from Asian and create another race class. In addition, Hawaiian, Pacific Islanders, and Native Americans are removed due to the insufficient number of the subject in the base images/videos. Consequently, there are five races in HiDF: White, Black, Asian, Latino, and Indian. We use skin color measurements from the Individual Typology Angle (ITA) (Wilkes et al., 2015) to minimize the subjectivity of the annotator in racial classification. Age is labelled as one of three categories (child, middle-aged adult, and elderly) as specified by Dammak et al. (2021).

**Annotation process** The annotation process for three categories (i.e., race, gender, and age) was conducted by three annotators. Instead of one-shot annotation that performs the annotation task for the whole data once, we iterate the process of (i) annotation for 1 K sample images for the three categories, (ii) measuring Cohen's kappa score for each category's annotations, and (iii) adjusting the annotation criteria through discussion when the score was below 0.8. The iteration process ends until Cohen's kappa score exceeded 0.8 for all categories among three annotators. Note that a kappa score of 0.8 or above indicates a high level of agreement among the annotators, as reported in the prior studies (Warrens, 2015; Liu et al., 2020a; Roth et al., 2020). The final kappa scores for race, gender, and age were 0.832, 0.970, and 0.875, respectively, indicating a high level of agreement (see Appendix A).

### 3.1.3 FAKE DATA GENERATION

**Deepfake generation tool** With the advancement of deepfake generation technology, various commercial tools (e.g., Reface, ZAO, FakeApp) have emerged, allowing the public to easily create deepfake images or videos (Masood et al., 2021). Among these, we selected Reface (Reface) as the deepfake generation tool for this study, considering factors such as service availability, the

convenience of the user interface (UI) design, the time required to generate deepfake images and videos, and the cost of using the service, which are all suitable for large-scale data generation (See Appendix B). Reface is one of the most accessible tools for the public and has recently gained significant popularity (Dang & Nguyen, 2023; Nawaz et al., 2023; Masood et al., 2023; Mehta et al., 2023; Nawaz et al., 2022). It employs a generation method based on Generative Adversarial Network (GAN) (Oles Petriv, 2021), producing visually natural results. Additionally, Reface can generate both deepfake images and videos, and its short generation time makes it suitable for creating a large volume of deepfakes.

**Deepfake generation process**   We generate a large number of deepfake images and videos by recruiting 50 paid applicants. The weekly goal of the applicant is to generate 700 and 210 fake images and videos, respectively. To achieve this, the randomly selected 700 base images, 840 target images, and 210 base videos are given to each applicant every week. By putting a pair of a base image and a target image to Reface, the applicant can obtain the synthesized image. The applicant is also required to verify the quality of the generated image. If the quality is low, another target image is used for retry. The whole process is repeated until a high-quality fake image is obtained. Similarly, the pair of a base video and a target image is used to generate a fake video. The amount of the weekly payment of each applicant is $50.6. The collection task was conducted for 74 days (from March 22 to June 3 in 2024). Approximately, $6,580 was spent in total.

**Data quality inspection**   Instead of using all the generated images and videos, we conduct a manual inspection to ensure that the generated content is human-indistinguishable. In particular, we exclude the generated fake images within one of the following criteria: (i) the positions of the eyes, nose, or mouth deviated significantly from their ideal locations or showed distortions (e.g., warping, blurring), (ii) the synthesized face overlapped with other body parts (e.g., the mouth is composited onto the back of the hand when the mouth was covered with the hand). For fake videos, we use the first criterion for fake images and additionally two more criteria: (i) synthesized eyes twitched (ii) lip or teeth movements were unnatural during speech (e.g., teeth protruding beyond the lips, lips not moving). Detailed examples of cases excluded during the quality inspection process are provided in Appendix B. After the quality check, we finally obtain 30,250 fake images and 4,241 videos in total.

## 3.2 DATASET DESCRIPTION

Throughout the construction process, we finally obtain 29,856 fake images and 4,241 fake videos, as shown in Table 1. Since 6,217 celebrities in CelebA-HQ and non-celebrities in FFHQ are included in base images and videos of HiDF, we ensure that the number of subjects should be more than 6,217 although we cannot compute the exact number as the number of the subject in FFHQ is unknown. The number of subjects in fake content can not be counted exactly since it depends on the number of subjects used for generation of fake images and videos. Considering that the number of fake images and videos, we estimate around 3.1 K and 1.6 K subjects may appear in fake images and videos, respectively, in expectation. Note that the number of subjects is much higher than the ones of other datasets.

The race, gender, and age distributions of the subjects constituting the target images utilized in the construction of HiDF are presented in Appendix C. First, in the race category, images and videos consisting of individuals with white ethnicity represent the majority, at 74.6% and 80.5%, respectively. In the case of images, Asian follows at 10.2%, while for videos, Latino comes next at 8.8%. Indian is the least represented in both images and videos, with 1.8% and 1.6%, respectively. The high proportion of White subjects is attributed to the CelebA-HQ and FFHQ datasets, from which the target images were extracted, exhibiting a similar demographic bias. Regarding gender, women constitute 62.1% and 61.1% of the images and videos, respectively, outnumbering men. For ages, adults comprise 83.1% and 90.0% of the images and videos, respectively. Children were unlikely to be contained due to the violation of criteria during the selection of target images. For example, there are more number of subjects eating or covering their faces with their hands. In addition, we found that a significant proportion of elder people with excessive wrinkles or sagging facial features was omitted because their synthesized results tended to appear unnatural. Consequently, the proportions of children and the elderly are relatively low.

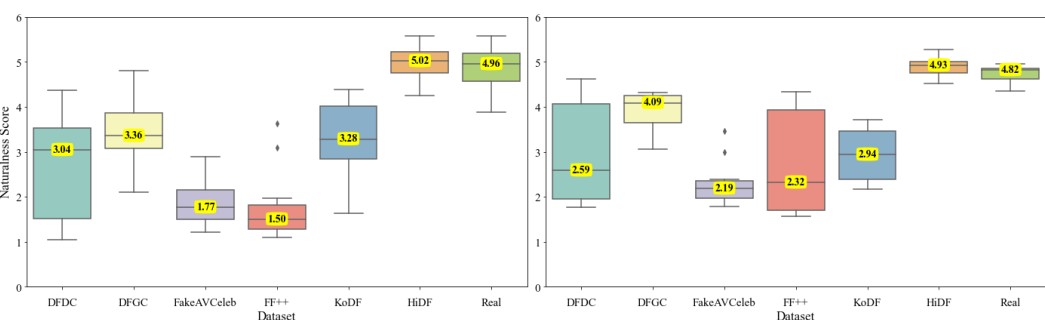

Figure 3: Overall qualitative results. Left: image, right: video.

Table 2: Results of quantitative quality assessment.

| Metric | FakeAVCeleb | DFDC | FF++ | KoDF | DFGC | HiDF |
|---|---|---|---|---|---|---|
| FID↓ (Heusel et al., 2017) | 22.971 | 23.516 | 18.440 | 29.512 | 35.695 | **13.005** |
| FVD↓ (Unterthiner et al., 2018) | 294.257 | 335.350 | 284.770 | 314.788 | 439.006 | **271.346** |

**Ethical considerations** HiDF was generated and constructed based on two image datasets and two video datasets. All the datasets used here are publicly available and confirmed to be permitted for redistribution and modification. The HiDF dataset is available under the Creative Commons Attribution-NonCommercial 4.0 International Public License https://creativecommons.org/licenses/by-nc/4.0/.

## 4 EXPERIMENTS AND RESULTS

### 4.1 QUALITATIVE DATASET ASSESSMENT

**Experimental settings** To qualitatively assess the data quality of HiDF, we conduced a survey study to evaluate the naturalness of deepfake images and videos. Crowdsourcing can result in age group bias, leading to unfair evaluation results. Thus, we categorized participants by age group, ranging from teenagers to individuals in their 50s, ensuring at least 10 participants per age group. Sixty-eight participants were recruited for the survey, with detailed participant demographics described in Appendix D. Participants were compensated approximately $11 upon survey completion; a total wage of $742 was used. Each participant evaluated 210 items, including 140 images and 70 videos generated using deepfake technologies. The image and video data were sourced from DFDC, DFGC, FakeAVCeleb, FF++, KoDF, HiDF, and original data. The original data consisted of real data from the five deepfake datasets, excluding HiDF, with 20 images and 10 videos randomly selected from each dataset. Participants rated the naturalness of the deepfake content on a scale from 1 (very unnatural) to 6 (very natural). To avoid neutral responses and elicit more definitive positive or negative opinions, we exclude the 'neutral' option from the 7-point Likert scale (Joshi et al., 2015). The detailed survey questionnaire is described in Appendix E.

**Results** The qualitative evaluation results for HiDF are summarized in Figure 3. Both the image and video results demonstrate that the perceived naturalness of HiDF is significantly higher than that of existing deepfake datasets. DFDC and KoDF were observed to include a wide range of data, from highly unnatural deepfake images and videos to relatively natural-looking ones. While FF++ images appear unnatural when viewed as single frames, some videos appear relatively natural when viewed as moving sequences. In contrast, HiDF consistently exhibits a high level of naturalness, often exceeding that of the original data. These results demonstrate that HiDF, having undergone rigorous inspection, comprises high-quality data that are indistinguishable by humans.

### 4.2 QUANTITATIVE DATASET ASSESSMENT

**Experimental settings** We conducted a quantitative quality assessment using Fréchet Inception Distance (FID) (Heusel et al., 2017) and Fréchet Video Distance (FVD) (Unterthiner et al., 2018),

Table 3: Overall performance.

| Type | Baseline | Year | FakeAVCeleb | | DFDC | | FF++ | |
|------|----------|------|-----|------|-----|------|-----|------|
| | | | AP↑ | AUC↑ | AP↑ | AUC↑ | AP↑ | AUC↑ |
| video | AVAD | 2023 | 0.939 | 0.837 | 0.717 | 0.656 | - | - |
| video | MARLIN-L | 2023 | 0.683 | 0.635 | 0.878 | 0.878 | 0.743 | 0.700 |
| video | MARLIN-B | 2023 | 0.638 | 0.578 | 0.827 | 0.844 | 0.682 | 0.659 |
| video | MARLIN-S | 2023 | 0.587 | 0.542 | 0.835 | 0.854 | 0.737 | 0.683 |
| video | FTCN | 2021 | 0.921 | 0.808 | 0.759 | 0.746 | - | - |
| video | EB4+EB4ST+B4Att+B4AST | 2020 | 0.937 | 0.830 | 0.888 | 0.876 | 0.940 | 0.925 |
| image | MARLIN-L | 2023 | 0.671 | 0.637 | 0.830 | 0.829 | 0.764 | 0.708 |
| image | MARLIN-B | 2023 | 0.678 | 0.617 | 0.634 | 0.660 | 0.640 | 0.661 |
| image | MARLIN-S | 2023 | 0.642 | 0.613 | 0.746 | 0.754 | 0.745 | 0.695 |
| image | EB4+EB4ST+B4Att+B4AST | 2020 | 0.930 | 0.815 | 0.871 | 0.861 | 0.914 | 0.898 |

| Type | Baseline | Year | KoDF | | DFGC | | HiDF | |
|------|----------|------|-----|------|-----|------|-----|------|
| | | | AP↑ | AUC↑ | AP↑ | AUC↑ | AP↑ | AUC↑ |
| video | AVAD | 2023 | 0.610 | 0.510 | 0.751 | 0.671 | **0.510** | **0.456** |
| video | MARLIN-L | 2023 | 0.513 | 0.513 | 0.709 | 0.973 | **0.511** | **0.491** |
| video | MARLIN-B | 2023 | 0.523 | 0.523 | 0.894 | 0.902 | **0.538** | **0.492** |
| video | MARLIN-S | 2023 | 0.51 | 0.51 | 0.908 | 0.920 | **0.507** | **0.483** |
| video | FTCN | 2021 | 0.914 | 0.897 | 0.864 | 0.808 | **0.631** | **0.697** |
| video | EB4+EB4ST+B4Att+B4AST | 2020 | 0.903 | 0.857 | 0.925 | 0.890 | **0.712** | **0.733** |
| image | MARLIN-L | 2023 | 0.890 | 0.883 | 0.928 | 0.931 | **0.530** | **0.528** |
| image | MARLIN-B | 2023 | 0.932 | 0.922 | 0.868 | 0.879 | **0.498** | **0.497** |
| image | MARLIN-S | 2023 | 0.879 | 0.875 | 0.918 | 0.926 | **0.498** | **0.492** |
| image | EB4+EB4ST+B4Att+B4AST | 2020 | 0.913 | 0.879 | 0.916 | 0.879 | **0.722** | **0.697** |

which are widely used quantitative metrics for evaluating the quality of synthesized data. FID is a standard metric used to evaluate GANs, measuring the Fréchet distance between the feature spaces of the generated image set and the real image set by calculating their means and covariances. FVD extends this concept of FID to video data. Lower scores in both FID and FVD indicate a closer distance between the real and generated distributions, implying that the generated images look natural. We utilized the implemented code to perform the calculations for FID and FVD in our experiments (Seitzer, 2020; calculate FVD, 2023).

**Results** In Table 2, DFGC shows the most significant difference between the synthesized results and the originals compared to other datasets, with FID and FVD scores of 35.695 and 439.006, respectively. DFDC employs seven post-processing methods to create more visually natural results, indicating extensive manipulation throughout the images. Thus, while they may appear natural to the human eye (see Figure 3), their similarity to the original images decreases when comparing pixel-level distributions. A lower similarity to the original images implies that the feature vectors extracted from real and fake images are not similar. In other words, it indicates that the distinction between real and fake is clear. While people perceive the DFGC images as natural in Figure 3(left), the deepfake detection rate for DFGC is generally high, as shown in Table 3. FF++ generally received low scores in qualitative assessment; however, in quantitative results, FID and FVD scores of 18.440 and 284.770 indicated relatively small differences between the synthesized results and the originals. FF++ used early deepfake generation methods (e.g., FaceSwap (Faceswap, 2018), Face2Face (Thies et al., 2016)), which swap faces according to a specified mask size. When only the designated area is synthesized, there is no harmony between the synthesized part and the base image, resulting in visually unnatural outcomes. However, at the pixel level, the information outside the synthesized part remains the same as the original.

In contrast, HiDF shows the highest data consistency with FID and FVD scores of 13.005 and 271.346, respectively. Given that HiDF recorded the highest scores in qualitative results, it is possible to achieve natural synthesis results while preserving the characteristics of the original images. Furthermore, the high similarity between real and fake images in HiDF means that it is difficult to distinguish between real and fake, as confirmed by the low detection performance in Table 3, which will be detailed later. These results suggest a future research direction on deepfake detection with human-indistinguishable deepfake data.

Table 4: Cross-dataset evaluation results. The dataset before the slash represents the one used for training and validation, while the dataset after the slash represents the one used for testing.

| Type | Baseline | HiDF / HiDF | | HiDF / DFGC | | DFGC / DFGC | | DFGC / HiDF | |
|---|---|---|---|---|---|---|---|---|---|
| | | AP↑ | AUC↑ | AP↑ | AUC↑ | AP↑ | AUC↑ | AP↑ | AUC↑ |
| video | MARLIN-L | 0.511 | 0.491 | 0.472 | 0.446 | 0.709 | 0.973 | 0.489 | 0.498 |
| video | MARLIN-B | 0.538 | 0.492 | 0.498 | 0.489 | 0.894 | 0.902 | 0.503 | 0.501 |
| video | MARLIN-S | 0.507 | 0.483 | 0.473 | 0.465 | 0.908 | 0.920 | 0.493 | 0.499 |
| image | MARLIN-L | 0.530 | 0.528 | 0.512 | 0.517 | 0.928 | 0.931 | 0.487 | 0.475 |
| image | MARLIN-B | 0.498 | 0.497 | 0.515 | 0.522 | 0.868 | 0.879 | 0.482 | 0.484 |
| image | MARLIN-S | 0.498 | 0.492 | 0.500 | 0.475 | 0.918 | 0.926 | 0.515 | 0.496 |

## 4.3 PERFORMANCE COMPARISONS WITH POPULAR DEEPFAKE METHODS

**Experimental settings** We next conduct performance comparisons with six popular deepfake detection baselines: AVAD (Feng et al., 2023), MARLIN-L, MARLIN-B, MARLIN-S (Cai et al., 2023), FTCN (Zheng et al., 2021), and EB4+EB4ST+B4Att+B4AST (EB4) (Bonettini et al., 2021). When selecting baselines, we prioritized deepfake detection methods with high detection rates and official code releases. The baselines used for evaluating deepfake detection performance and detailed parameter settings are described in Appendix D. Note that the experiments were conducted separately for deepfake image detection and video detection. AVAD and FTCN are deepfake video models that utilize visual and audio modalities. Since FF++ does not include audio, its performance on this dataset was omitted.

**Results** In both the deepfake video and image detection experiments, all the detection methods exhibit significantly lower performance on HiDF than those with other existing datasets (See Table 3). While the performance on existing deepfake datasets varies depending on the structure of the baseline and the type of pre-trained datasets used, detecting deepfake images and videos from HiDF tend to be difficult in general. This indicates that, compared to existing deepfake datasets where the difference between real and fake is relatively straightforward, the deepfake images and videos in HiDF are more challenging to distinguish from real ones, demonstrating their high quality.

## 4.4 CROSS-DATASET EVALUATION

**Experimental settings** To assess the effectiveness of HiDF in deepfake detection, we conducted a cross-dataset evaluation. Specifically, we compared the performance of models trained on HiDF against those trained on the DFGC dataset, which includes various manipulation techniques. The DFGC dataset is created using eight synthesizing methods (DeepFaceLab (Perov et al., 2020), SimSwap (Chen et al., 2020), FaceShifter (Li et al., 2019), FaceSwapper (Li et al., 2024), MegaFS (Zhu et al., 2021), InfoSwap (Gao et al., 2021), Self-proposed method (Peng et al., 2021), and YouTube-DF (Kukanov et al., 2020)) and four commercial tools (ZAO, FaceMagic (FaceMagic), Reface (Reface), Jiggy (Jiggy)), covering a range of manipulation types. We evaluated deepfake detection performance on images and videos across the following four conditions: (1) training and testing with HiDF, (2) training with HiDF and testing on DFGC, (3) training and testing with DFGC, and (4) training with DFGC and testing on HiDF. For the cross-dataset evaluation, we used MARLIN-L/B/S baselines, with the data split into train, validation, and test sets in a 6:2:2 ratio.

**Results** Table 4 shows that when MARLIN-L is trained on HiDF and tested on DFGC, it achieves an AP of 0.473 and an AUC of 0.446. Conversely, when trained on DFGC and tested on HiDF, the AP and AUC are 0.489 and 0.499, respectively. The performance difference between these two cases is marginal, with just 0.016 and 0.053 differences. A similar pattern is observed in the other baselines. These results indicate that HiDF, despite being generated with fewer manipulations than other deepfake datasets, can play a similar role with other deepfake datasets with various manipulations. Additionally, the notable performance gap when trained and tested on DFGC compared to training on DFGC and testing on HiDF highlights the need for further investigation into human-indistinguishable datasets like HiDF.

## 5 DISCUSSION AND LIMITATIONS

**Data availability and social impact**  In this study, we proposed and built a human-indistinguishable high-quality deepfake image and video dataset, and will publicly release HiDF to advance deepfake detection research. HiDF has great utility across various research areas -— from developing new algorithms for deepfake generation to experimenting with and evaluating the efficiency of deepfake detection systems. In this way, we expect it to contribute significantly to the progress and advancement of emerging deepfake technology research. However, deepfake technology poses various risks at both individual and societal levels. By emphasizing the importance of personal data protection, HiDF aims to raise awareness about the potential misuse of deepfakes and contribute to enhancing societal awareness of these risks.

**Limitations**  Despite the significant contributions of HiDF, we clearly indicate the limitations. First, all annotations conducted for deepfake data generation aimed at producing natural results, thereby failing to encompass various conditions such as instances where multiple faces appear or when facial features are heavily obscured. This could be addressed by leveraging more advanced deepfake generation techniques in the future.

Second, relying on a single commercial tool for deepfake generation can limit the ability to detect a variety of synthesis methods. However, as shown in Table 4, HiDF offers a comparable performance with other deepfake datasets that used diverse synthesis techniques. While new synthesis methods are continually emerging in academia, it takes time for them to be adopted commercially (Rogers et al., 2014). As a result, publicly available deepfake tools often represent older methods. Additionally, many widely used commercial tools no longer offer support or require significant payment for high-quality deepfake creation, restricting access to the general public. Given these limitations, we selected the most accessible tool to create HiDF. This ensures that HiDF reflects the type of deepfake data commonly generated by the public, making it a valuable resource for practical deepfake detection.

Lastly, the race, gender, and age distribution of the subjects in HiDF may exhibit potential bias. However, since fine-grained labels are provided for each category, diverse applications in various contexts can be possible. For example, researchers can evaluate models in diverse scenarios tailored to their ongoing research needs, such as attempting verification with data excluding specific racial groups to check for biases in developed models. Furthermore, detailed labeling of data enhances transparency and clarity, enabling the identification of categories with insufficient data and facilitating efficient data augmentation.

## 6 CONCLUSION

In this paper, we introduce HiDF, a novel high-quality and human-indistinguishable deepfake dataset, which comprises 30 K deepfake images and 4 K deepfake videos. We meticulously select the base and target images, and base videos, which enable the most natural deepfake generation through thorough and comprehensive annotation. We generated the data using a commercial deepfake generation tool and ensured high quality through rigorous post-screening. We validated the superior quality of HiDF compared to the existing datasets through quantitative and qualitative assessments. We further compared the performance of popular deepfake detection models on HiDF and existing datasets, demonstrating the need for further research on indistinguishable data. We expect HiDF to support practical deepfake detection tasks and serve as a valuable benchmark dataset.

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

## A  APPENDIX

The following document is the supplementary material for our paper, **HiDF: A Human-Indistinguishable Deepfake Dataset**. First, Section A details the selection process for base images, target images, and target videos, and describes the consistency results of annotations for race, gender, and age. Section B describes the tools used for generating images and videos in HiDF, and the reviewing process of the images and videos that are finally included in HiDF. Section C provides the number of images and videos in HiDF, categorized by race, gender, and age, and then explains the methods for maintaining and managing HiDF. Section D analyzes the participants' demographics in the survey conducted for the Qualitative Quality Assessment for evaluating HiDF. Also, the baselines used in the performance comparison between HiDF and other deepfake datasets, along with the various parameter settings, are briefly described. Finally, Section E presents the survey configuration used in the Qualitative Quality Assessment.

## A  PREPROCESSING FOR BUILDING HiDF

In this paper, when swapping the face of image A with that of image B, we refer to image A as the base image and the image to be swapped (i.e., image B) as the target image. We select base and target images and videos based on the predifined criteria to generate natural deepfake images and videos. Section A.1 provides the detailed criteria for selecting base images and target images along with examples, while Section A.2 presents the criteria for selecting base videos.

To support a broader range of applications in deepfake detection research, we conduct annotations in terms of race, gender, and age. Section A.3 analyzes the proportions of race, gender, and age for the subjects included in the final selected base images, target images, and base videos.

### A.1  CRITERIA FOR SELECTING BASE AND TARGET IMAGES

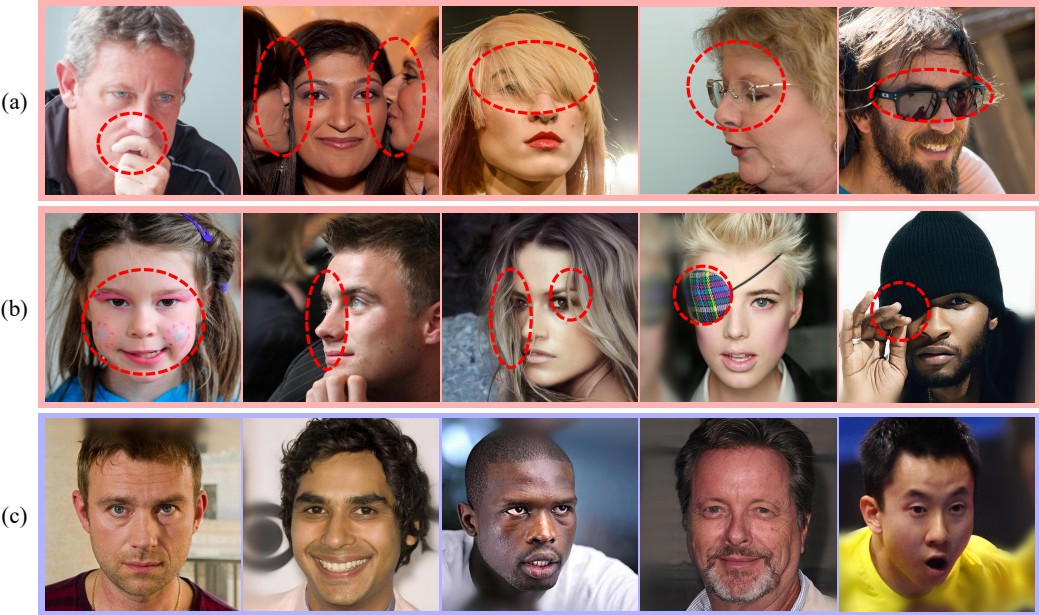

Figure 4: Examples of base and target image selection criteria. (a) Examples that do not meet the criteria for base or target images. (b) Examples that are suitable only as target images. (c) Suitable examples for both base and target images.

To select visually natural deepfake images, we choose base and target images based on the criteria detailed in Section 3.1.1. Figure 4a illustrates cases that do not meet the conditions both for base and target images, with detailed reasons as follows (from left to right). In the leftmost image, if the mouth or nose is covered by a hand, it limits obtaining information about those parts. The next

image containing more than one person can cause confusion when detecting the face to be swapped. In the third image, if more than half of the face is obscured, accurate face extraction is impossible. Next, a side-facing face makes it difficult to obtain information about features such as the eyes, and if accessories obscure the face, accurate facial recognition is challenging. In the rightmost image, wearing opaque accessories such as sunglasses prevents obtaining information about the eyes.

Next, cases that do not meet the conditions for base images but can be selected as target images are shown in Figure 4b. In the leftmost image, a face with makeup, such as face painting, can appear unnatural if used as a base image, but they can be used as a target image as long as no parts of the face are obscured and sufficient information can be extracted. In the next image, a side-facing face without obscuring accessories allows for obtaining the face's shape from the visible side. For the third image, if a part of the face is obscured by hair, using it as a base image can result in a blurred synthesis of the hair, but it can be used as a target image if the general outline is visible. For the next two images, if an accessory obscures one eye, face information can still be obtained from the other side, similar to the second image case.

Lastly, Figure 4c showcases examples that are ideal both for base and target images. These are cases where the face is facing forward, and the facial features are clearly visible, providing the best conditions for deepfake image creation.

## A.2 CRITERIA AND EXAMPLES FOR SELECTING BASE VIDEOS

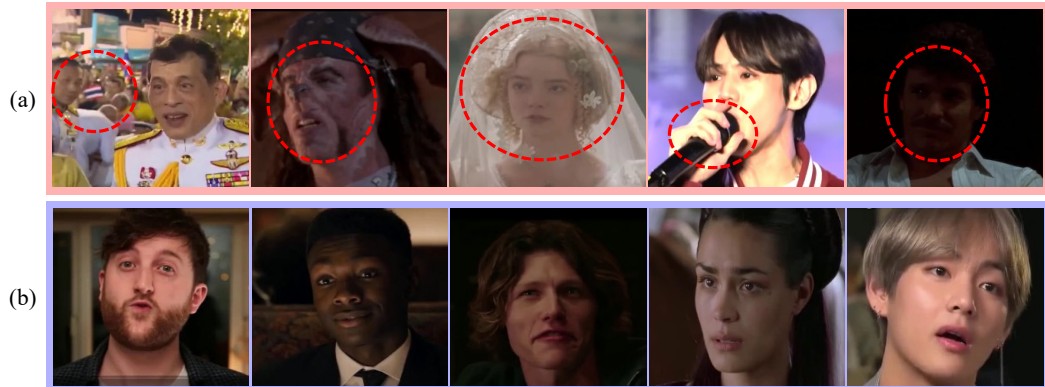

Figure 5: Examples of inappropriate and appropriate base videos. (a) Inappropriate cases for base videos. (b) Appropriate cases for base videos.

Inappropriate cases for base videos are illustrated in Figure 5a. In the leftmost image, as with base and target images, videos featuring multiple people are excluded. Next, the base videos used to construct HiDF include clips from dramas and movies uploaded to YouTube. Due to this, many instances tend to involve computer graphics (CG). Artificially altered facial features hinder the creation of natural deepfake results, so these cases are excluded. The third video where the face is obscured by accessories, such as a veil, tends to prevent facial recognition. Next, when a person's face is partially obscured by an object, such as holding a microphone while speaking, the synthetic results of face swap in these areas are unnatural. In the rightmost video, if the lighting is too low, it becomes challenging to accurately recognize the face's position, decreasing the likelihood of placing the target face correctly.

Additionally, videos where the subject is not speaking are also excluded. More specifically, videos in which the voice belongs to a third party off-screen or only background music is present fall under this category. The effectiveness of multimodal information in deepfake detection lies in capturing subtle mismatches between the manipulated face and the actual speech. However, if the subject is not speaking, the audio information cannot be effectively utilized, necessitating this additional condition. This process ensures the composition of videos where audio-visual information corresponds.

Figure 5b shows appropriate examples of base videos. Similar to the criteria for suitable images, the subject in the video should primarily face forward, with clear and distinguishable facial features. Additionally, there should be minimal facial and body movement throughout the video.

Table 5: Details of total images, base and swap images, and base videos for each dataset.

| Type | Category | CelebA-HQ | FFHQ | FakeAVCeleb | YouTube |
|---|---|---|---|---|---|
| Image | # Selection | 24,700 | 25,900 | - | - |
| | # Base | 17,980 | 16,953 | - | - |
| | # Target | 20,648 | 21,129 | - | - |
| Video | # Selection | - | - | 500 | 15,400 |
| | # Base | - | - | 453 | 5,254 |

Table 1 summarizes the number of base and target images and videos selected for the construction of HiDF. To avoid an imbalance in the usage of a single image dataset, we selected approximately 25 K images from CelebA-HQ and FFHQ for annotation. Due to the more varied environments in which FFHQ images were captured, more of these images were excluded during the selection of base images compared to CelebA-HQ. For videos, we annotated all the real videos from FakeAVCeleb and 15,400 videos collected from YouTube. Since the YouTube videos featured considerable movement and were filmed under diverse lighting conditions, a significant number were excluded during the selection of base videos.

### A.3 Inter-annotator agreements on race, gender, and age annotations

Table 6: Cohen's kappa score between annotators A1, A2, and A3 for race, gender, and age categories.

| Category | A1&A2 | A2&A3 | A1&A3 | Mean |
|---|---|---|---|---|
| Race | 0.867 | 0.809 | 0.822 | **0.832** |
| Gender | 0.984 | 0.972 | 0.955 | **0.970** |
| Age | 0.920 | 0.863 | 0.844 | **0.875** |

To support more robust deepfake detection performance, HiDF includes fine-grained labeling of subjects' races, genders, and ages. This detailed annotation is essential for various applications in deepfake detection research, such as evaluating performance for specific races or improving deepfake detection for older individuals. Therefore, we annotated each target image with the following labels: race, gender, and age.

Three annotators were involved in the meticulous annotation process. Prior to annotating the entire set of target images, we undertook a rigorous process to enhance the agreement among annotators for each category (see Section 3.1.2), ensuring the reliability of our results. Table 6 presents Cohen's kappa scores for 1,000 target images, which shows the annotators' agreement. The kappa scores exceed 0.8 for race, gender, and age categories, demonstrating high consistency among the annotators.

## B Data Generation and Description

### B.1 Data generation tool

As of June 10th, 2024, the most accessible commercial tools for creating deepfakes for the general public are Reface[2], FakeApp[3], and ZAO (See Section 3.1). Reface was finally chosen for generating the deepfake images and videos that constitute HiDF due to its suitability for large-scale data generation. This section compares Reface and FakeApp in terms of cost, deepfake generation time, and interface design. Note that ZAO is excluded as the service was not available at the moment.

When it comes to cost, Reface is a budget-friendly option for large-scale data generation, offering unlimited face swaps for a reasonable cost, $29.99, per month. In contrast, FakeApp is free but requires a minimum of 8 GB RAM and high-performance GPU hardware. In terms of deepfake generation time, Reface delivers results within a quick time, 10 seconds, for images, and around 30 seconds for videos. In contrast, FakeApp can take from several hours to days, depending on hardware

---

[2]Reface. `https://reface.ai/unboring/face-swap`.
[3]FakeApp 2.2.0. `https://www.malavida.com/en/soft/fakeapp/`.

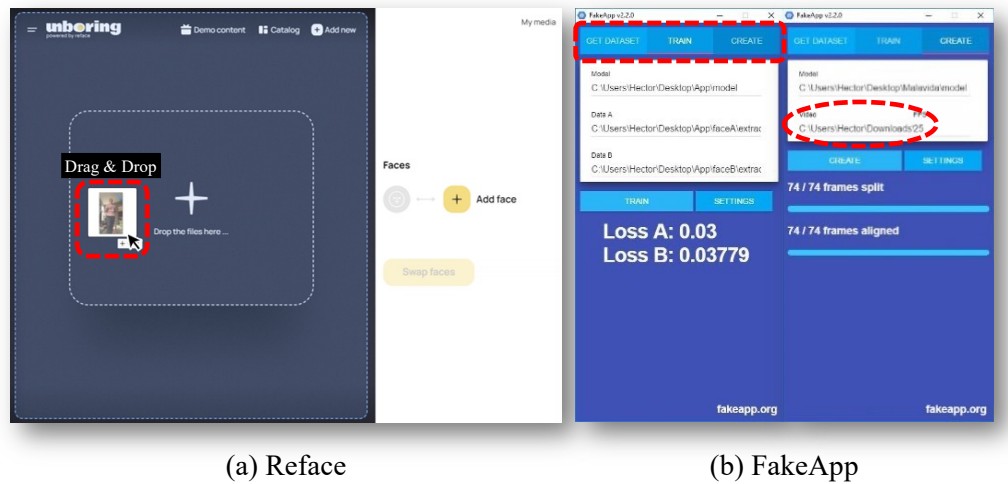

|                | (a) Reface | (b) FakeApp |

Figure 6: Comparison of Commercial Deepfake Tools. (a) Reface Interface. (b) FakeApp Interface.

performance and the size of the training dataset. Moreover, as illustrated in Figure 6a, Reface's interface is designed to be intuitive and user-friendly, making it a breeze for users to select and upload desired images or videos and generate results with a single click. On the other hand, Figure 6b shows that FakeApp is more complex, involving three stages: 'Get Dataset,' 'Train,' and 'Create.' It also requires users to input folder paths for the image and video samples used for face swaps, which can be a bit cumbersome and time-consuming.

Given that generating deepfake data using commercial tools generally demands considerable time, Reface is more suitable for large-scale deepfake data generation than other commercial tools due to its cost efficiency, rapid generation time, and user-friendly interface.

## B.2 DATA QUALITY INSPECTION

To construct a high-quality, human-indistinguishable deepfake dataset, we manually inspected and filtered the generated deepfake images and videos based on the following criteria (See Section 3.1.3). Figure 7a shows examples of deepfake images excluded during this inspection process, with the following detailed reasons (from left to right). First, images where the eyes, nose, and mouth are misaligned are excluded. Second, we excluded the images with visible hairline boundaries due to improper removal of the target face's hair. Third, the images with abnormal deformations in specific facial features are excluded. Fourth, we excluded instances where the eyebrows of the base image are obscured, leading to unnatural synthesis of the target face's eyebrows. Fifth, the images where the target face's eyebrows are covered by hair, resulting in distorted eyebrow features, were excluded.

Figure 7c presents examples of deepfake videos that were excluded. In the first image, cases where a hand or other body part obscures the face, causing the target image to overlap improperly, were excluded. Second, we excluded the videos with visible hairline boundaries. Third, the videos with a significant discrepancy between the facial shapes of the base and target faces, causing sync issues and unnatural results, were excluded. Fourth, we excluded instances where the subject turns their face sideways, leading to improper synthesis and partial exposure of the base face. Lastly, the videos where the target face's eyebrows are covered by hair were excluded.

Figures 7b and 7d show the deepfake images and videos that were ultimately included in HiDF. These carefully selected examples do not exhibit the issues identified in Figures 7a and 7c. They appear natural and seamless, providing high authenticity to human observers.

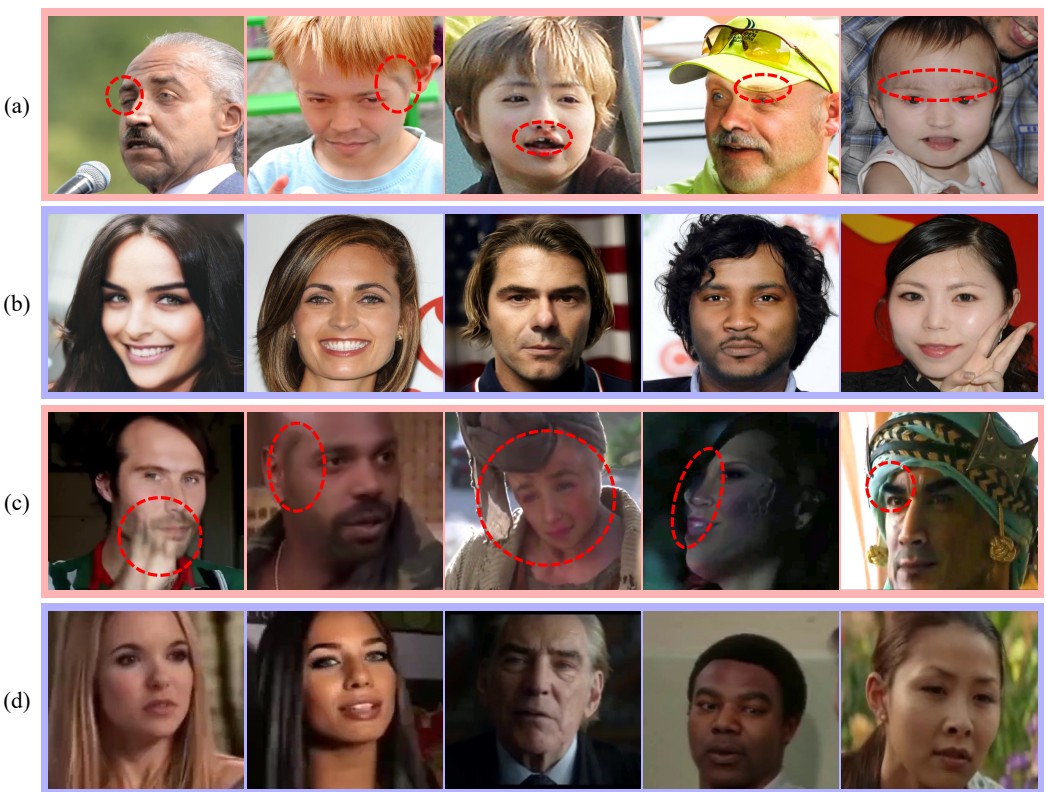

Figure 7: Examples of deepfake image and video inspection criteria. (a) Examples of excluded deepfake images. (b) Examples of included deepfake images. (c) Examples of excluded deepfake videos. (d) Examples of included deepfake videos.

Table 7: Statistics of our dataset. All values are denoted as percentages.

| Type | Race | | | | | Gender | | Age | | |
|------|------|------|------|------|------|------|------|------|------|------|
| | White | Black | Asian | Latino | Indian | Male | Female | Child | Adult | Elderly |
| Image | 74.6 | 5.9 | 10.2 | 7.5 | 1.8 | 37.9 | 62.1 | 9.6 | 83.1 | 7.3 |
| Video | 80.5 | 5.3 | 3.8 | 8.8 | 1.6 | 38.9 | 61.1 | 1.5 | 90.0 | 8.5 |
| Total | 74.9 | 5.8 | 9.8 | 7.7 | 1.8 | 39.7 | 60.3 | 9.2 | 83.4 | 7.4 |

## C  DATA DESCRIPTION

### C.1  DATA DESCRIPTION BY RACE, GENDER, AND AGE

Table 7 summarizes the demographic analysis of characters appearing in deepfake images and videos in HiDF. The racial distribution in deepfake images shows that Caucasians account for the highest proportion at 74.6%, followed by Asians at 10.2%, Latinos at 7.5%, Blacks at 5.9%, and Indians at 1.8%. Similarly, in deepfake videos, Caucasians represent the highest proportion at 80.5%, followed by Latinos at 8.8%, Blacks at 5.3%, Asians at 3.8%, and Indians at 1.6%, showing a slightly different pattern compared to images. Both deepfake images and videos include more females than males, and the age group primarily consists of adults (83.4%), with lower proportions of children and elderly individuals. CelebA-HQ and FFHQ datasets used for generating HiDF exhibit a slight bias towards specific races (i.e., white) and age groups (i.e., adults), resulting in distributions similar to those shown in Table 7.

### C.2  DATA PUBLICATION AND LICENSING

We open HiDF publicly available at `https://github.will.be.provided`. The HiDF dataset is available under the Creative Commons Attribution-NonCommercial 4.0 International Public License `https://creativecommons.org/licenses/by-nc/4.0/`.

## D  EXPERIMENTAL RESULTS

### D.1  QUALITATIVE DATASET ASSESSMENT

Table 8: Demographics of survey participants.

| Category | Subcategory | | | | |
|---|---|---|---|---|---|
| **Gender** | **Male** | **Female** | | | |
| # of participant | 41 | 28 | | | |
| **Age** | **10s** | **20s** | **30s** | **40s** | **50s** |
| # of participant | 12 | 21 | 13 | 10 | 13 |
| **Occupation** | **College** | **Graduate** | **Employees** | **Entrepreneurs** | **Unemployed** |
| # of participant | 21 | 14 | 23 | 4 | 7 |

To perform a qualitative dataset assessment of HiDF, we surveyed 69 participants from diverse backgrounds (See Table 8). Among the participants, the gender distribution includes 60% male and 40% female, with the 20s age group being the most represented, comprising 21 individuals. We ensured a minimum of 10 participants from the five age groups ranging from teens to 50s, with the 40s being the least represented. This approach aimed to secure sufficient responses across all age groups, thereby enhancing the reliability and representativeness of the assessment results. The participants' occupations were categorized into five groups (i.e., college students, graduate students, employees, entrepreneurs, and unemployed), with college students making up the largest group of 21 participants.

We analyzed the survey results, which assessed the naturalness of five deepfake datasets (i.e., FakeAVCeleb(Khalid et al., 2021), DFDC(Dolhansky et al., 2020), FF++(Rossler et al., 2019), KoDF(Kwon et al., 2021), DFGC(Peng et al., 2021)), HiDF, and original images and videos by gender and age group. Figures 8 and 9 present the gender-based analysis, while Figures 10 and 11 show the age-based analysis. Overall, the qualitative dataset assessment results indicate that HiDF images and videos received significantly high scores in all cases. Notably, HiDF scored as high as or higher than the original images and videos, demonstrating that HiDF is composed of human-indistinguishable deepfake data. Additionally, HiDF exhibited a much narrower interquartile range (IQR) than other datasets. These findings suggest that samples from the HiDF dataset were consistently perceived as highly natural across all age and gender groups.

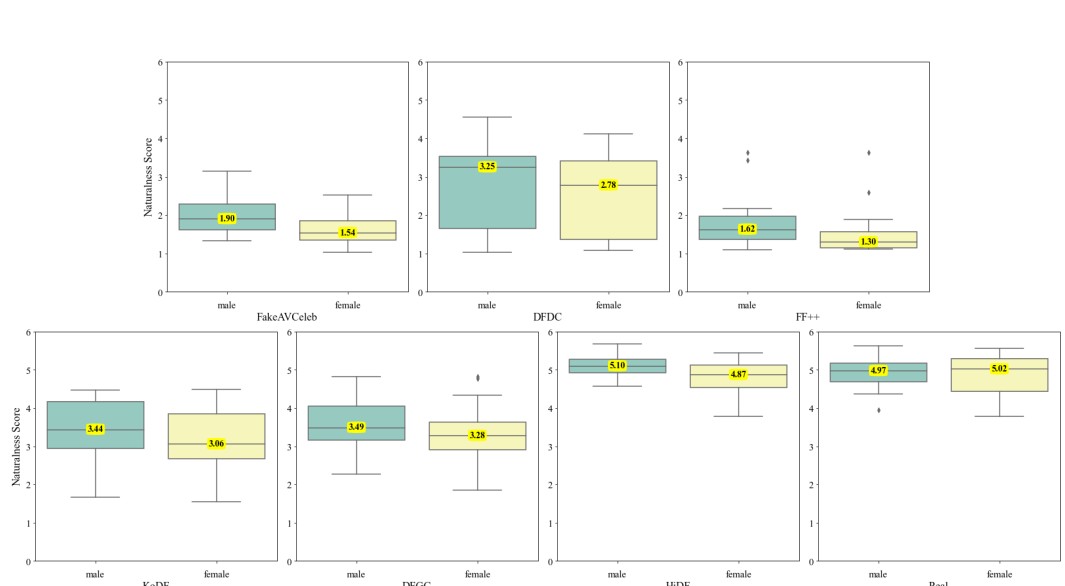

Figure 8: Qualitative assessment results on images by gender.

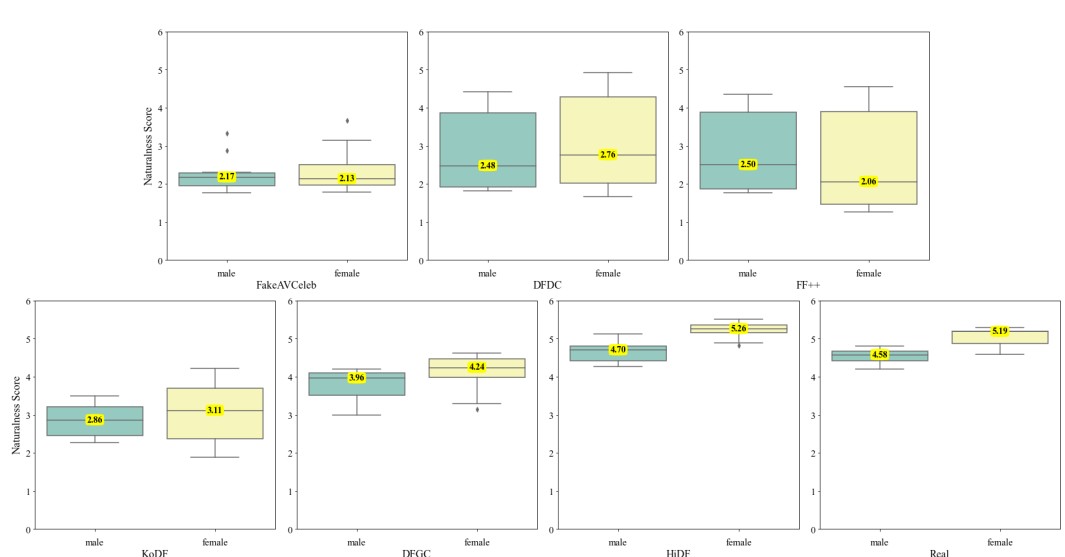

Figure 9: Qualitative assessment results on videos by gender.

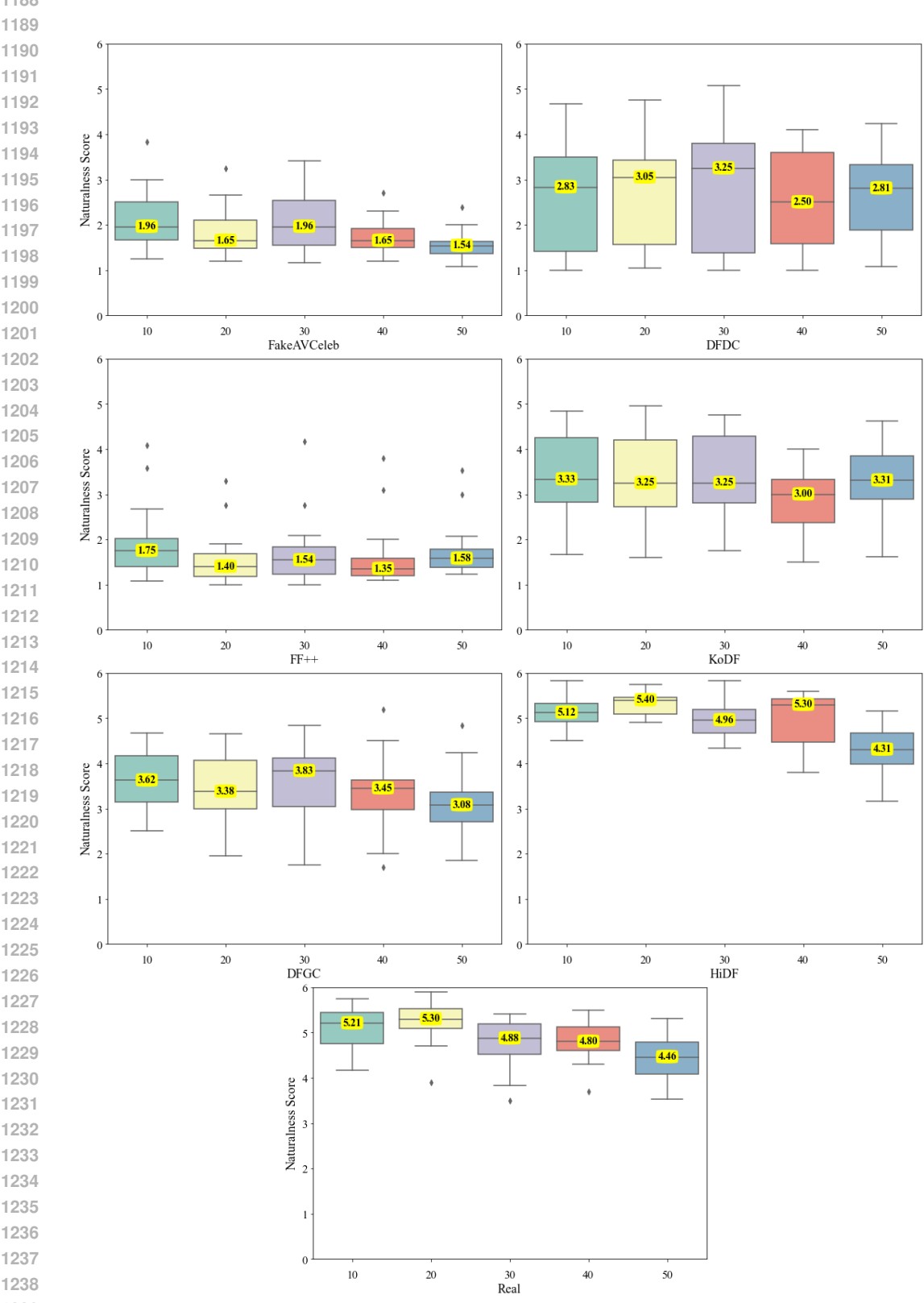

Figure 10: Qualitative assessment results on images by age.

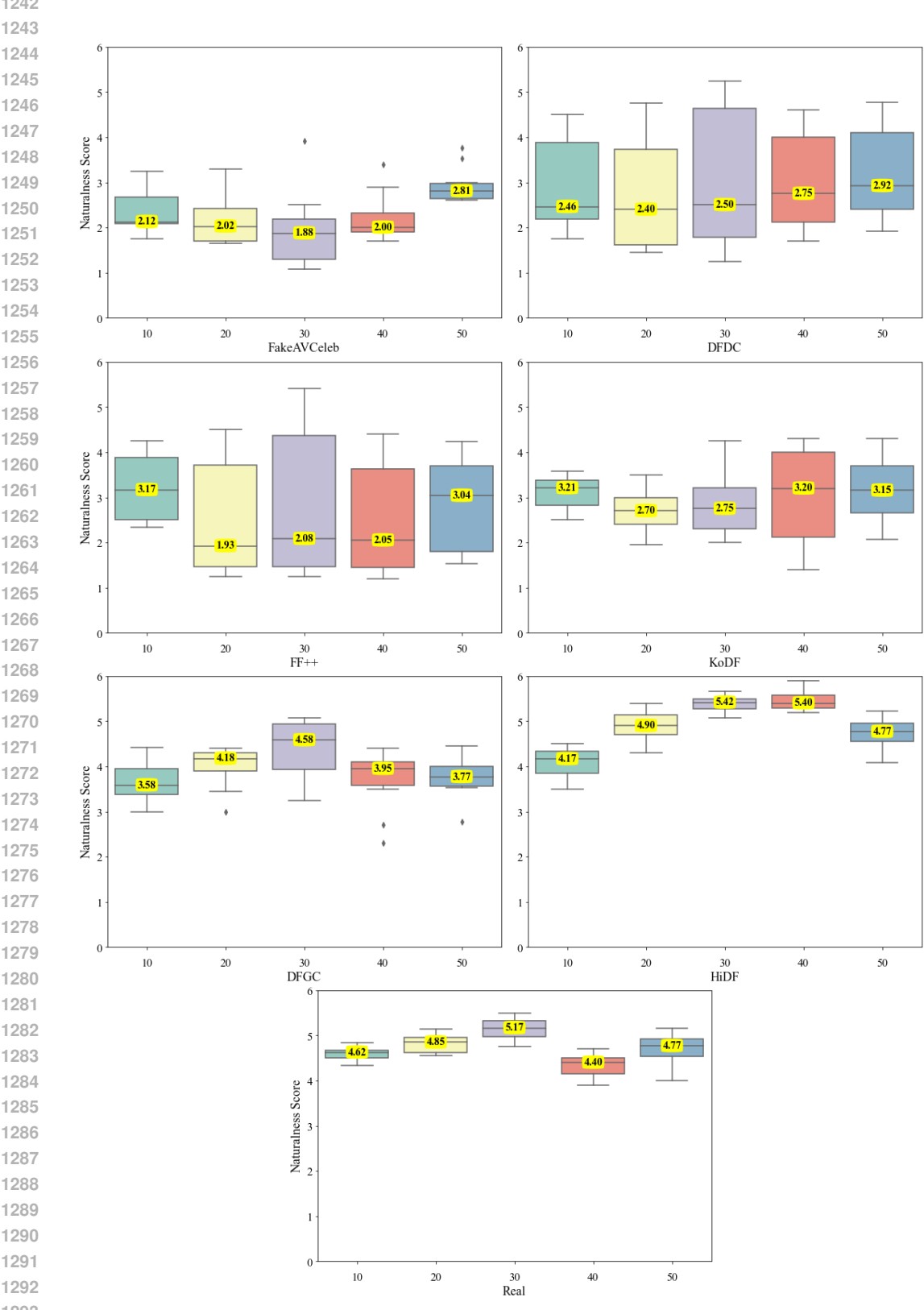

Figure 11: Qualitative assessment results on videos by age.

### D.2 BASELINES FOR PERFORMANCE COMPARISONS

Below is a brief description of the baselines of deepfake image and video detection methods used to compare the detection performance of HiDF with other deepfake datasets in our experiments. For the deepfake image and video detection task, we primarily selected methods that have released official code among the state-of-the-art (SoTA) and top-performing methods.

- **MARLIN**(Cai et al., 2023) extracts universal facial representations via self-supervised learning, applicable to various computer vision tasks, and demonstrates excellent performance in deepfake detection tasks. We applied MARLIN in this deepfake detection task, distinguished into small, base, and large models based on weight size, suitable for both image and video datasets.
- **EfficientNetB4 + EfficientNetB4ST + B4Att + B4AttST**(Bonettini et al., 2021) is an ensemble of various CNN models to detect deepfakes. This architecture combines attention layers and Siamese training across two distinct base networks (EfficientNet and B4), achieving superior performance through ensemble methods compared to individual models. This method is applicable to both images and videos.
- **AVAD**(Feng et al., 2023) is a method that is trained on real video data to effectively detect discrepancies between visual and auditory signals in manipulated videos. It captures temporal synchronization features in videos to generate continuous audio-visual features using autoregressive models, which are models that can predict future values based on past values, enabling the detection of abnormal patterns.
- **FTCN**(Zheng et al., 2021) utilizes temporal coherence to detect manipulated faces. Composed of an end-to-end model with a fully temporal convolution network for extracting temporal features and a Temporal Transformer network for considering long-term temporal coherence, FTCN is adept at identifying manipulated faces over time.

### D.3 EXPERIMENTAL SETTINGS

**A strategy for constructing a test set.** To compare deepfake image detection performance, we construct the image dataset used in the experiments by extracting frames from each deepfake video dataset. For FakeAVCeleb, which contains only 500 real videos, we randomly extracted a single frame from each of the 500 real videos and 1500 fake videos. For the other datasets, 1000 real and 1000 fake videos were randomly selected, and one frame was extracted from each, resulting in a dataset of 2000 images. Similarly, 1000 deepfake images from HiDF were selected, and the real images were composed by randomly selecting 500 images each from the two datasets used for generating HiDF's deepfake images (i.e., CelebA-HQ and FFHQ). For video datasets, except for FakeAVCeleb, which consists of 500 real videos and 1500 fake videos, we randomly selected 1000 real videos and 1000 fake videos from each dataset, resulting in a total of 2000 videos used for the experiments. The HiDF deepfake videos were selected in the same manner, while the HiDF real video set comprised 200 videos from FakeAVCeleb's real videos and 800 videos collected from YouTube. Both images and videos underwent a face cropping process to focus on the facial regions.

**Parameter settings.** For the deepfake video detection baseline models, we used the AVAD model with the official code settings. The FTCN model had its threshold set to 0.04, based on the settings from the AltFreezing(**?**) that uses FTCN. For the EfficientNetB4 + EfficientNetB4ST + B4Att + B4AttST model, which can detect both deepfake images and videos, we followed the official code settings for parameters such as image size and frame count. We only used a model pretrained on DFDC when evaluating the test set extracted from DFDC. For the other datasets (i.e., FakeAVCeleb, FF++, KoDF, DFGC, HiDF), we used models pretrained on FF++. The MARLIN model, primarily designed for extracting facial features, required an additional simple deepfake classification model. We used an SVM model with an RBF kernel for classification, adjusting the gamma values to 0.003, 0.005, and 0.008 for MARLIN large, base, and small, respectively. For training the classification model, each dataset was split into train, validation, and test sets in a 6:2:2 ratio, ensuring balanced class proportions using the stratify parameter.

**Computing resources and evaluation metrics.** All experiments were conducted on NVIDIA GeForce 631 RTX 3090 and NVIDIA CUDA GPUs. We use AUC (Area Under Curve) and AP (Average Precision) as evaluation metrics for all models.

# E  SURVEY QUESTIONNAIRE

This section presents the content of the survey designed for qualitative dataset assessment. In Figure 12, the first page of the survey introduces the purpose, expected time commitment, and evaluation methods. Following this, as shown in Figure 13, the survey is structured to obtain consent for future information usage from participants and collect necessary personal information. Specific qualitative assessment scores, as illustrated in Figures 14 and 16, are categorized into six levels ranging from 'Very Unnatural' (1) to 'Very Natural' (6). Figure 16, focusing on evaluating deepfake videos, includes fields for necessary information regarding assessing these videos. Due to technical constraints preventing video uploads directly to the survey form, videos were uploaded to Google Drive (See Figure 18) with links provided for participants to access. Figures 15 and 17 depict sample survey items.

# [Online Survey] Evaluation of the Naturalness of Deepfake Dataset

**B**  *I*  U  ⊖  𝕏̶

Hello,

We are the DSAIL(Data Science & Artificial Intelligence) Lab at Sungkyunkwan University. We are conducting a **quality assessment** of dataset generated using **deepfake** technology.

**[ Details ]**

- Duration : 30 minutes

**[ Evaluation Method ]**

This survey aims to evaluate the naturalness of images and videos generated using deepfake technology.

Deepfakes are created by synthesizing another person's face onto the original image or video. We are interested in hearing your opinions on **how natural** these composites appear.

When evaluating,
rather than judging whether **"every detail, such as the surface of the face, looks smooth and natural"** (since it is stated that deepfake technology was applied), please focus on assessing **"whether these images/videos would be convincing enough not to raise suspicion if seen on the internet."**

During surveys, you will be shown various deepfake images and video clips. (Unmanipulated images/videos are also included.)

Please rate each item on a scale from **"Very Unnatural (1)"** to **"Very Natural (6)."**

Your valuable opinion will greatly contribute to the advancement of deepfake technology and ethical useage.

**[ Contact ]**

Sungkyunkwan University DSAIL Jonghyun Lee

Tel:

Email:

Figure 12: Survey overview.

Participant information and survey results from this study will be used for research purposes. Other personal information will only be used for compensation payment, and all participant data will be deleted after payment. This information will not be disclosed to external parties, and the information used in the research cannot identify participants. Do you agree to the collected information being used in other studies for better research in the future? (If you do not consent, you cannot participate in the experiment.)

◯ Yes, Agree

◯ No, Disagree

Please select your age group.

◯ 10s (10 to 19 years old)

◯ 20s (20 to 29 years old)

◯ 30s (30 to 39 years old)

◯ 40s (40 to 49 years old)

◯ 50s (50 to 59 years old)

Please select your gender.

◯ Male

◯ Female

Please enter your occupation. (e.g. student, office worker, etc.)

answer

Please enter your name.

answer

Please enter your mobile phone number.
(e.g., 010-1234-5678 / To request information for future compensation)

answer

Figure 13: Participants' consent and information collection.

# [Online Survey] Evaluation of the Naturalness of Deepfake Dataset

email address

⊠ 비공개

## 1. Deepfake image quality assessment

Please rate the naturalness of the deepfake image on a scale from 1 to 6.

- 1: Very Unnatural
- 2: Unnatural
- 3: Little Unnatural
- 4: Little Natural
- 5: Natural
- 6: Very Natural

Figure 14: Rating scale for deepfake images.

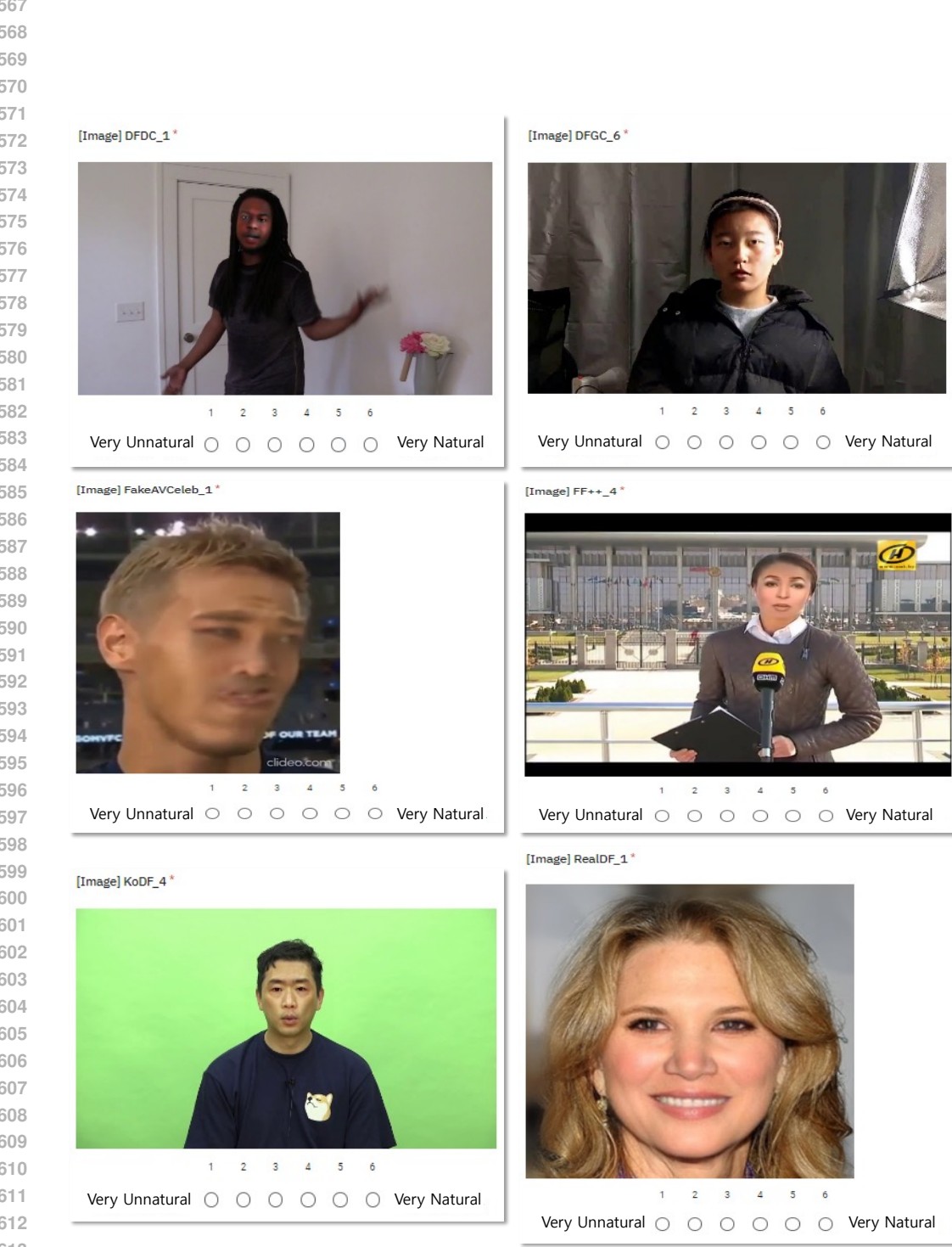

Figure 15: Examples of deepfake image evaluation items.

3 중 3 섹션

## 2. Deepfake video quality assessment

Please rate the naturalness of the deepfake video on a scale from 1 to 6.

You can check the at the link below.
Video Link

**[ Issue ]**

- Problem
  : When a Windows OS user downloads a video, the **video does not open** after number 30.

- Solution
  : Instead of downloading the videos, you can directly double-click on the files within Google Drive (on the web) to view them.

Please evaluate the quality of the video corresponding to its video name.

- 1: Very Unnatural
- 2: Unnatural
- 3: Little Unnatural
- 4: Little Natural
- 5: Natural
- 6: Very Natural

Figure 16: Information for assessing deepfake videos.

Figure 17: Examples of deepfake video evaluation items.

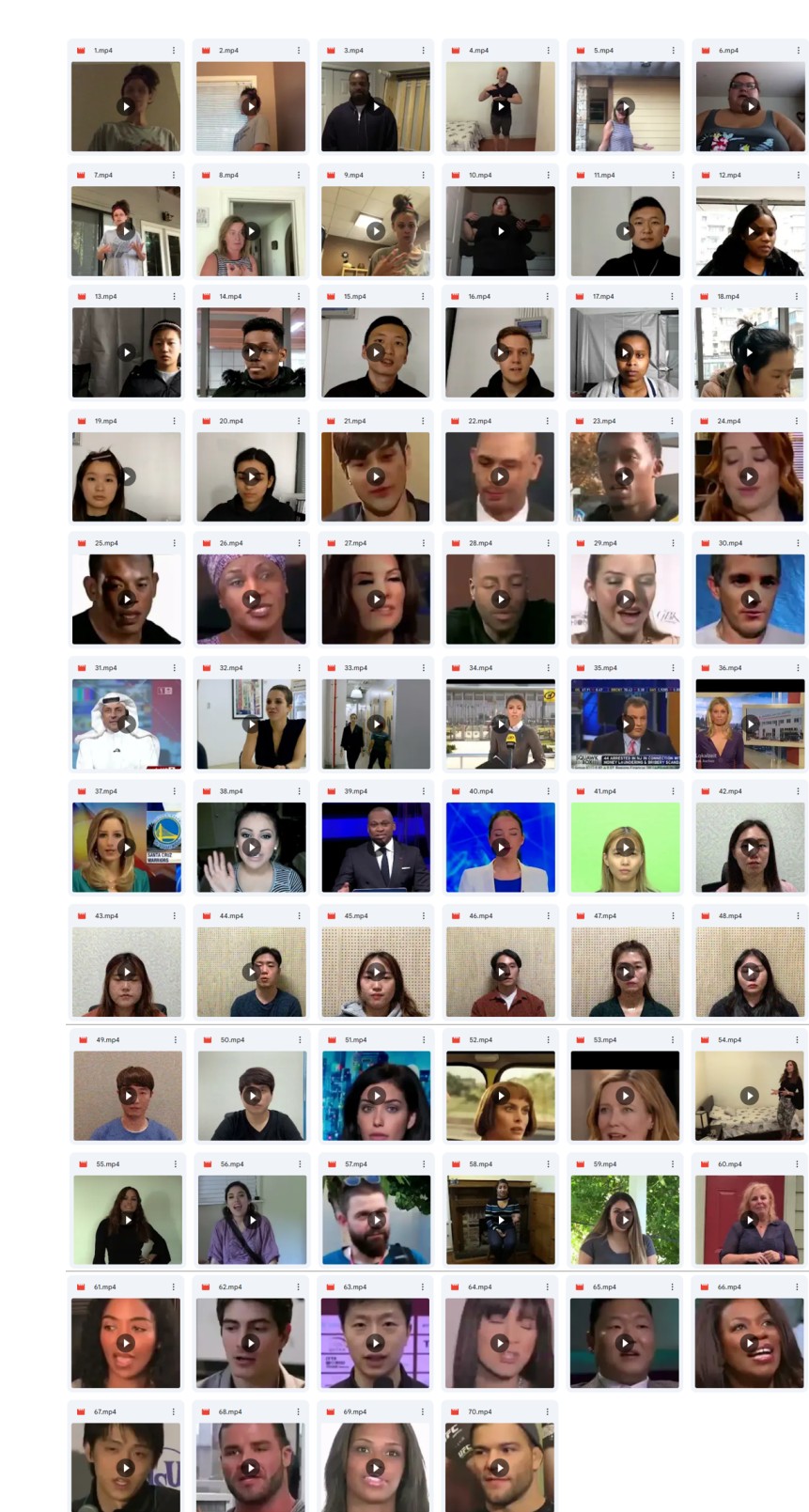

Figure 18: Video samples used for evaluation.

