# OpenReview forum: "HiDF: A Human-Indistinguishable Deepfake Dataset"
_ICLR.cc/2025/Conference — ICLR 2025 Conference Withdrawn Submission_

### Official Review · Reviewer_8n37 · 2024-11-01

**Soundness:** 2
**Presentation:** 3
**Contribution:** 2
**Rating:** 5
**Confidence:** 4

**Summary:**

This paper proposes HiDF, a high-quality deepfake dataset that is perceptually indistinguishable from real data. The collection and annotation of the dataset is described, qualitative and quantitative experiments are conducted to verify its high-quality, and deepfake detection evaluations are carried out on this dataset and others.

**Strengths:**

The continuous contribution of new dataset to the deepfake detection area is important, and this is the main contribution of this work.
The quality of the HiDF dataset is high and verified by human evaluations. It is created using an advanced commercial deepfake software, and the base and target images and videos are diverse and carefully selected.
Clear descriptions of the dataset are presented to draw more attention to use it in future research.

**Weaknesses:**

Although the proposed dataset is of high quality, only one tool is used to generate all the data, making the dataset not diverse in quality and generation method. A deepfake detection model trained on HiDF may not be able to detect low-quality deepfakes, limiting its effectiveness in training detection models.

The contribution is somewhat limited by only proposing a new dataset. (Line 097)

The experimental evaluation on deepfake detection is relatively weak. In Table 3, some detection methods are evaluated, but no method is effective enough on HiDF (the highest AUC is just around 0.7). It is recommended to test some more recent and more powerful models. If no method is effective, some analysis of the cause should be presented and discussed.

Some factual mistakes about the dataset DFGC should be corrected. Around Line 184, the commercial tools are used to generate the private-2 test set which is not released, and dataset actually contains 2,799 fake videos (should exclude the private-2 set). Please refer to the dataset webpage https://github.com/NiCE-X/DFGC-2022?tab=readme-ov-file#the-detection-dataset

**Questions:**

Some small questions:
What is the \alpha stands for in Table 1?
Around Line 187, “the videos are still human-distinguishable since the co-provided audio is not related to the subjects in the video” I do not understand and thinks this statement may be inaccurate, since the audios in the DFGC dataset are kept from the original base videos.
A row of average performance over detection methods can better present the results in Table 3.
What is the reference real data for each deepfake dataset in calculating FID and FVD in Section 4.2?
Line 413 it should be DFGC instead of DFDC?
The description of the subjects distribution is too verbose at Line 312-323. Showing figures is more efficient that the words.

---

### Official Review · Reviewer_LDNQ · 2024-11-01

**Soundness:** 2
**Presentation:** 3
**Contribution:** 2
**Rating:** 3
**Confidence:** 5

**Summary:**

The paper presents Human-Indistinguishable DeepFake (HiDF), a novel, high-quality deepfake dataset comprising 30,000 images and 4,000 videos, curated to ensure indistinguishability from real content by human observers. HiDF is designed to fill a critical gap in deepfake research by offering a challenging benchmark that reflects realistic, high-fidelity visual content generated using a commercial tool (Reface). Each item in HiDF includes demographic annotations, such as race, gender, and age, enabling deeper exploration into biases and fairness in model performance.

**Strengths:**

+ High-quality dataset and rigorous validation

+ HiDF’s creation involved quality control, with specific exclusion criteria for obstructed or unnatural images and videos, ensuring consistently high visual quality across the dataset.

+ HiDF includes a notable range of subjects with varying racial, gender, and age attributes. Despite this diversity, the dataset leans heavily toward white subjects due to its dependence on CelebA-HQ and FFHQ, introducing potential bias.

+ The dataset and code are released under a CC license.

+ HiDF offers synthetic content that closely aligns with deepfake styles accessible to the public, enhancing its relevance for real-world applications.

**Weaknesses:**

- The primary concern is the dataset’s reliance on a single-generation method (Reface), which restricts its applicability to diverse real-world deepfake scenarios. This limitation hinders the dataset’s utility as a comprehensive benchmark.

- HiDF predominantly includes images and videos with clear, unobstructed facial views, which may not reflect real-world conditions where faces are often obscured by objects like hair, hats, or hands (also explored in [1]). This idealized selection reduces the dataset’s applicability for training models to detect deepfakes in more complex, everyday scenarios, potentially leading to poorer model performance in real-world detection tasks.

- While HiDF contains various demographic attributes, some reviews noted a potential imbalance in race, gender, and age distributions. Such imbalances could introduce bias into detection models trained on HiDF.

- It is moderate relative to some large-scale datasets in the field. A larger sample size could improve the dataset’s robustness and efficacy as a deepfake detection benchmark.

- The paper could strengthen its contributions by including qualitative evaluations using additional metrics, such as the BRISQUE score, to provide a deeper visual quality assessment. Moreover, baseline comparisons using multi-modal and unimodal methods, such as MRDF and FACTOR for audio-visual evaluations and ASSIST or RAwGAT-ST for audio-specific baselines, would provide insights into the dataset’s performance across detection modalities. Showing generalization capability for baseline models trained on HiDF would further clarify the dataset’s efficacy in varied detection scenarios.

References:
[1] Narayan et al., DF-Platter: Multi-Face Heterogeneous Deepfake Dataset, CVPR 2023.

**Questions:**

Please address the weaknesses.

---

### Official Review · Reviewer_uMme · 2024-11-03

**Soundness:** 1
**Presentation:** 2
**Contribution:** 1
**Rating:** 3
**Confidence:** 5

**Summary:**

This paper presents a deepfake dataset, HiDF, consisting of 30,000 real images, 4,000 real videos, and corresponding forged objects. The motivation for its creation is that existing datasets for deepfake detection are predominantly distinguishable to the naked eye and exhibit significant discrepancies from real-world forged objects. This disparity hampers the application of deepfake detection models in real-world scenarios and hinders their development. To address this issue, this paper constructs a high-quality and human-indistinguishable deepfake dataset using Reface, a widely used deepfake tool in real-world applications. This dataset aims to facilitate the advancement of deepfake detection models and supports the evaluation of their deployment capabilities in real-world contexts.

**Strengths:**

● This paper aims to improve the quality of existing deepfake videos and constructs a higher-quality deepfake dataset using the commercial software Reface.
● To ensure high quality, sufficient data preprocessing is carried out in advance, and the generated data is strictly reviewed and evaluated using both qualitative and quantitative methods.
● Several experiments are conducted to demonstrate the proposed dataset is natural, and even better than real videos.

**Weaknesses:**

● The link given in the abstract is not working. If the dataset is ready, I suggest providing the ``real'' link here. If not,  adding a  ``pseudo'' link here is not helpful.

● The contribution appears limited. This dataset was created using an existing commercial tool with some curated engineering operations to filter out failed samples, but the demonstration experiments do not fully support the stated claims (see Questions box). While I recognize that building a higher-quality dataset could benefit the deepfake detection community, the forgery technique used is restricted to Reface, resulting in low diversity among generated faces. I suspect that even if a method performs well on the proposed dataset, its effectiveness could degrade significantly in real-world applications. Increasing diversity through various forgery techniques is essential for creating robust benchmarks. Additionally, achieving “human-indistinguishable” quality is not the primary challenge in deepfake datasets. Instead, the focus should be on accurately simulating the real-world distribution of deepfake faces. However, the paper does not seem to provide sufficient explanations or experimental support for this aspect. Finally, I would expect improvements in deepfake generation techniques, such as a pipeline that dynamically switches gates, parameters, or structures within a single model to produce a variety of forgery types. Merely using curated engineering pipelines with limited technical innovation may fall short of ICLR’s standards.

● This paper does not mention the comparison of the proposed dataset with another high-quality dataset Celeb-DF [CVPR 2020]

● Another major concern is that only using a single commercial tool (Reface) introduces highly limited diversity. Even though the visual quality is good, this dataset still hardly covers the distribution of deepfake faces in real-world scenarios. (see a related question in the Questions box)

● In experiments, the baseline detection methods are outdated and limited in number, only including AVAD (Feng et al., 2023), MARLIN-L, MARLIN-B, MARLIN-S (Cai et al., 2023), FTCN (Zheng et al., 2021), and EB4+EB4ST+B4Att+B4AST (EB4) (Bonettini et al., 2021). Such evaluation is not comprehensive and hardly demonstrates the superiority of the proposed dataset. Many recent methods can be referred to, such as:
1. Implicit Identity Leakage: The Stumbling Block to Improving Deepfake Detection Generalization   CVPR2023（image）
2. LAA-Net: Localized Artifact Attention Network for Quality-Agnostic and Generalizable Deepfake Detection      CVPR 2024 (image)
3. Self-Supervised Video Forensics by Audio-Visual Anomaly Detection CVPR2023（video）
4. Leveraging Real Talking Faces via Self-Supervision for Robust Forgery Detection cvpr2022(video)

● There are a few spelling errors, such as in line 352 (conducted -> conducted), and in the appendix on line 1337, the citations are incomplete (AltFreezing(?)).

**Questions:**

1. In the cross-dataset experiments (Table 4), the MARLIN-L model achieve a detection AUC of 0.491 when trained and tested on the HIDF dataset. However, when trained on the DFGC dataset and tested on HIDF, one would expect a decline in performance (which is indeed the case for images). Surprisingly, the AUC for MARLIN-L in the cross-testing on the HIDF dataset was 0.498, even surpassing the previous result, and the same trend is observed for MARLIN-B and MARLIN-S. These results indicate that the DFGC dataset performs the worst in quantitative evaluations, revealing a significant pixel-level discrepancy from real videos. Even so, training a model on DFGC achieves impressive performance on the HIDF dataset, exceeding that of models trained directly on HIDF. One doubt is: if training on more comprehensive and extensive datasets—regardless of their inherent quality, with many videos being distinguishable to the naked eye—could still enable models to exhibit excellent generalization performance on the HIDF dataset? I highly suspect that the lack of forgery diversity results in this outcome and it highly hinders the necessity of this dataset.

2. The real videos in HIDF are composed of celebrities from YouTube and FakeAVCeleb. Given this, I am curious why the commonly used celebrity benchmark dataset in face forgery detection, Celeb-DF, is not utilized.

3. As I understand, the preprocessing stage for selecting images and videos, as well as the filtering stage after the generation of forged objects, aims at constructing realistic faces. So why can the selection criteria for target images be more restricted compared to source images? If the goal is to make the forgery process as perfect as possible, should not the filtering criteria for both types of images be the same?

---

> ### Comment · Reviewer_uMme · 2024-12-02
>
> Dear AC and reviewss, the authors have not provided responses for these questions. Thus i keep the original score.

---

### Official Review · Reviewer_RkAh · 2024-11-05

**Soundness:** 3
**Presentation:** 3
**Contribution:** 3
**Rating:** 6
**Confidence:** 2

**Summary:**

The authors have introduced the HiDF (Human Indistinguishable DeepFake Dataset) in this work. Previous DeepFake datasets are suffering from lower quality and can usually be distinguished by human. To build the new DeepFake dataset that can help developing the next-generation deepfake detector, the authors have made an effort to create the HiDF which contains 30K images and 4K videos.

The authors first select base and target images, as well as base videos and then generate the DeepFake data using commercial deepfake generation tool and ensure its quality by post-screening. Through quantitative and qualitative assessment, the authors have verified the superior quality of the HiDF dataset compared to existing datasets.

Baseline performance on popular deepfake detectors are conducted on this new datasets, demonstrating the need for further research on this more challenging deepfake dataset.

**Strengths:**

This paper presents a novel deepfake dataset that is human indistinguishable, a more challenging dataset for the entire deepfake detection community. Overall, the contribution is tangible and solid.

The authors also explain the curation process clearly in the paper, and they also showcased how the popular deepfake detection methods perform on this novel (more challenging) dataset.

**Weaknesses:**

1. the face swap is done using one commercial tool: Reface. (i) the dataset will absorb all the biases existed in the Reface tool. (ii) more diverse tools may be considered for face swap to make the dataset more diverse?

Alongside of Reface, maybe the authors can consider including 1-2 commercial or academic face swap tools, and discuss the rationale of including and excluding the tools. The authors are also encouraged to discuss the potential biases introduced by Reface and how they might impact the dataset's usefulness for deepfake detection research. Lastly, the authors are encouraged to discuss whether and how incorporating multiple tools could enhance the dataset's diversity and reduce potential biases.


2. only some deepfake datasets and detector methods are discussed and compared. I am not suggesting to add lots of baselines, but instead I am suggesting to include iconic ones from each category in terms of methodology, for example, one most-popular GAN-based method, one most-popular diffusion-based method, etc. See [1] and other recent survey for categorizations.

The authors are suggested to discuss and justify their choice of baseline methods and explain how they ensure comprehensive coverage of different deepfake detection approaches. The authors are also suggested to discuss whether and how a more diverse set of baselines might provide additional insights into the dataset's effectiveness.

[1] IJCV 2022 Countering Malicious DeepFakes: Survey, Battleground, and Horizon

**Questions:**

Please comment on the Weakness section and discuss these in the final version.

---

> ### Comment · Reviewer_RkAh · 2024-12-02
> **no author-initiated discussion**
>
> Dear ACs and fellow reviewers, since there is no author-initiated discussion, I am inclined to keep my initial rating.

---

### Note · Authors · 2024-12-03

I have read and agree with the venue's withdrawal policy on behalf of myself and my co-authors.